

# GAPNet: Single and multiplant leaf disease classification method based on simplified SqueezeNet for grape, apple and potato plants

Özge Nur Özaras[1] and Asuman Günay Yılmaz[2]

[1] Faculty of Technology, Department of Software Engineering, Karadeniz Technical University, Trabzon, Turkey
[2] Faculty of Engineering, Department of Artificial Intelligence and Data Engineering, Karadeniz Technical University, Trabzon, Turkey

## ABSTRACT

Humans need food to sustain their lives. Therefore, agriculture is one of the most important issues in nations. Agriculture also plays a major role in the economic development of countries by increasing economic income. Early diagnosis of plant diseases is crucial for agricultural productivity and continuity. Early disease detection directly impacts the quality and quantity of crops. For this reason, many studies have been carried out on plant leaf disease classification. In this study, a simple and effective leaf disease classification method was developed. Disease classification was performed using seven state-of-the-art pretrained convolutional neural network architectures: VGG16, ResNet50, SqueezeNet, Xception, ShuffleNet, DenseNet121 and MobileNetV2. A simplified SqueezeNet model, GAPNet, was subsequently proposed for grape, apple and potato leaf disease classification. GAPNet was designed to be a lightweight and fast model with 337.872 parameters. To address the data imbalance between classes, oversampling was carried out using the synthetic minority oversampling technique. The proposed model achieves accuracy rates of 99.72%, 99.53%, and 99.83% for grape, apple and potato leaf disease classification, respectively. A success rate of 99.64% was achieved in multiplant leaf disease classification when the grape, apple and potato datasets were combined. Compared with the state-of-the-art methods, the lightweight GAPNet model produces promising results for various plant species.

# INTRODUCTION

The rapid increase in the world population has led to a high demand for water and food resources. Owing to the scarcity and unequal distribution of these resources, approximately 720 million people faced hunger worldwide in 2020 (*Canton, 2021*). For this reason, it is essential to focus on food production processes to ensure healthy human life. Plant diseases can cause significant damage to crops, posing challenges to food shortages, sustainability, and food security (*Tiwari, 2021*). Early diagnosis of plant diseases ensures

Corresponding author
Özge Nur Özaras,
ozgenur.ozaras@ktu.edu.tr

the quality and continuity of agricultural productivity. Therefore, examining the stages of plant growth and development is crucial for detecting agricultural diseases and preventing crop losses. Additionally, the economic and social development of countries relies on agriculture (*Shewale, 2023*). Fast and accurate plant disease detection not only guarantees product quality and quantity but also increases economic income (*Yu, Xie & Huang, 2023*). Consequently, it is necessary to investigate plant leaf diseases and take early precautions to increase agricultural productivity.

Bacteria, fungi and viruses are the causes of plant diseases (*Özcan & Dönmez, 2021*). When these viruses infect a plant, they spread the disease to the plant's leaves and crops (*Babalola, Kpai & Toygar, 2023*). With the rapid development of technology, artificial intelligence and its subfields, machine learning and deep learning methods have gained attention in many research areas. Recently, many deep learning-based (convolutional neural network (CNN) and Vision Transformer (ViT)) studies have been carried out for the early detection of plant leaf diseases. Compared with CNN models, ViT models have satisfactory performance in image classification tasks, but they are relatively more complex and have high computational demands (*Rachman et al., 2024*). However, in real-world applications, it is not efficient to use models that require considerable memory and processing power. Therefore, there is a need to design models that will provide successful results in environments with few resources. The main objective of this study is to design a lightweight architecture that maintains high accuracy while significantly reducing computational demands. For this reason, a CNN-based model that is suitable for deployment on resource-constrained devices was developed.

The aim of this study is to design a CNN model suitable for deployment on resource-constrained devices used in agricultural environments. Additionally, this study aimed to develop a lightweight CNN model that performs well not only for one plant species but also for various plant species. The proposed GAPNet model uses an architectural approach based on the SqueezeNet model. It also performs disease classification on multiple plant species (grape, apple, and potato). Our study also provides a comprehensive comparative analysis against seven state-of-the-art CNN architectures. In this study, the leaf disease classification performances of seven pretrained CNN models, VGG16, ResNet50, SqueezeNet, Xception, ShuffleNet, DenseNet121, and MobilNetV2, were examined. Inspired by the SqueezeNet architecture, a lightweight CNN model, GAPNet, was proposed for grape, apple and potato leaf disease classification. To address the data imbalance problem between classes, the synthetic minority oversampling technique (SMOTE) (*Elreedy & Atiya, 2019*) method was used. SMOTE balances the dataset by generating synthetic examples for underrepresented classes, thus ensuring equal representation of classes and increasing the reliability of classification results (*Zaw & Mon, 2024*).

## Motivation for the study

Plant diseases can cause significant damage to crops, posing challenges to food shortages, sustainability, and food security (*Tiwari, 2021*). Therefore, examining the stages of plant growth and development is crucial for detecting agricultural diseases, taking precautions in

time, and preventing crop losses. Early detection of leaf diseases also helps decrease the use of chemical pesticides and thus decreases environmental and health risks. Since leaf diseases have different characteristics depending on the plant species, these diseases need to be recognized and classified correctly. Fast and accurate plant disease detection guarantees product quality and quantity. For this reason, plant leaf diseases should be investigated, and early precautions should be taken to increase agricultural productivity.

## Motivation for the choice of methods used

Recently, deep learning methods-CNNs, ViTs, *etc.*-have achieved high accuracy rates in the field of computer vision. These methods achieve superior performance to traditional machine learning methods by automatically learning image features. ViT models have satisfactory performance in image classification tasks, but they are relatively more complex than CNN models and have high computational demands (*Rachman et al., 2024*). For this reason, a CNN-based model that is suitable for deployment on resource-constrained devices was developed in this study. In this study, a lightweight and fast CNN model, GAPNet, is proposed for the early detection and classification of plant leaf diseases.

## Contributions

- Leaf disease classification was carried out for three different plant species (grape, apple and potato).
- Oversampling with the SMOTE method was applied to address the data imbalance problem.
- The leaf disease classification performance of seven state-of-the-art CNNs (VGG16, ResNet50, SqueezeNet, Xception, ShuffleNet, DenseNet121, and MobileNetV2) for three different plant species was evaluated.
- A fast and lightweight CNN model, GAPNet, was proposed for single and multiplant leaf disease classification.

## Structure of the manuscript

The article is organized as follows. The plant leaf disease classification studies in the literature are summarized in the next section. The dataset, the pretrained CNN models, and the proposed method are explained in section three. The fourth section presents experimental studies and the results. The final section provides a discussion and conclusions.

## Literature review

This section describes the literature on leaf disease classification. Over the years, this research topic has become more interesting, and many methods have been proposed in the literature for disease classification from plant leaves. *Nagaraju, Swetha & Stalin (2020)* proposed a fine-tuned VGG-16 network for the early detection and classification of apple and grape leaf diseases. Their method achieved 97.87% accuracy on a dataset consisting of grape and apple leaf diseases. *Çetiner (2021)* used pretrained DenseNet121, DenseNet201,

InceptionResNetV2, InceptionV3, and ResNet50V2 models for feature extraction and a CNN-based classification model for apple leaf disease detection and achieved the highest accuracy (99.00%) with ResNet50V2. *Reddy & Neeraja (2022)* proposed a system that combines DenseNet and 1D CNN models to detect plant leaf diseases. This system was used to classify diseases in apple, grape, potato, and strawberry leaf images and provide treatment recommendations. *Nagi & Tripathy (2022)* proposed a lightweight CNN model to classify grape leaf diseases. Their model outperformed the pretrained AlexNet, MobileNet, and VGG16 models, with 98.40% accuracy.

*Pradhan (2022)* used DenseNet201, DenseNet169, InceptionV3, InceptionResNetV2, MobileNet, MobileNetV2, ResNet50, VGG16, VGG19, and Xception models for apple leaf disease classification. DenseNet201 outperformed the other models with 98.75% accuracy. *Babalola, Kpai & Toygar (2023)* used AlexNet for apple leaf disease classification and achieved 99.56% accuracy. *Sood & Singh (2024)* achieved 99.88% accuracy with their CNN architecture for the early diagnosis of grape leaf diseases. *Sofuoğlu & Bırant (2024)* proposed a new CNN model for plant leaf disease classification. Their model achieved 98.28% accuracy on potato leaf images. *Upadhyay & Gupta (2024)* achieved an accuracy rate of 98.94% with the ResNeXt model for the early diagnosis of fungal diseases on apple leaves.

*Garma et al. (2022)* compared different state-of-the-art mobile CNN architectures for classifying maize leaf diseases and insect pests. The DiCENet, EfficientNet, GhostNet, MixNet, MobileNetV3, SPNASNet, FBNet, MNASNet, ShuffleNetV2 and SqueezeNext networks were selected because of their small size and lower computational complexity. The GhostNet model showed the highest performance, with an average accuracy of 97.78% in detecting plant leaf diseases. *Banjar et al. (2025)* proposed E-AppleNet, an advanced version of EfficientNetV2 that includes attention mechanisms for the classification of apple leaf diseases. With this model, they achieved 99% accuracy on the PlantVillage dataset.

*Doutoum & Tugrul (2025)* conducted a systematic analysis by investigating deep learning methods on datasets created for apple leaf disease detection and classification. They reported that deep learning techniques are more effective than traditional machine learning methods in leaf disease detection and classification. They also suggested using the apple dataset obtained from PlantVillage instead of datasets obtained from unstable environments. *Bonkra, Pathak & Kaur (2025)* developed a hybrid model that combines convolutional autoencoders (CAEs) and CNNs for apple leaf disease detection. They evaluated the performance of ResNet50, EfficientNetB3, and their hybrid model for apple leaf disease classification. They achieved 96% accuracy with the proposed hybrid CAE-CNN model. Another approach for classifying potato leaf diseases was presented by *Sarfarazi, Zefrehi & Toygar (2024)*. They used the fusion method of multiple color spaces to improve feature extraction and the weighted majority voting strategy to combine predictions from AlexNet, ResNet50, and MobileNet in the classification phase. As a result, the proposed approach achieved 98.61% accuracy on the PlantVillage dataset and 97.78% accuracy on the potato leaf dataset. *Kunduracioglu & Pacal (2024)* used a combination of 14 CNN and 17 image transform models for disease classification from grape leaves. The

four models achieved 100% accuracy on the PlantVillage and Grapevine datasets, highlighting the performance of the Swinv2-Base model.

*Özaras, Yilmaz & Gedikli (2024)* proposed the AppleSENet model for apple leaf disease classification. The squeeze and excitation (SE) blocks were used to generate channel attention weights, thus increasing the performance by highlighting the features that are effective in solving the problem. Compared with the SqueezeNet, ShuffleNet, and MobileNetV2 networks, AppleSENet achieved high classification performance (99.21%). While our previous work (*Özaras, Yilmaz & Gedikli, 2024*) focused solely on apple leaf disease classification *via* the SE Network approach, this study presents several important improvements: disease classification for multiple plant species (grape, apple, and potato), a new architectural approach based on a simplified SqueezeNet model, and a comprehensive comparative analysis against seven state-of-the-art CNN architectures.

## MATERIALS AND METHODS

In this study, pretrained CNN networks VGG16, ResNet50, SqueezeNet, Xception, ShuffleNet, DenseNet121, and MobileNetV2 were used to classify diseases from apple, grape and potato leaf images. The SMOTE method was applied to address the data imbalance problem in the datasets. Inspired by the SqueezeNet model, the lightweight GAPNet network was proposed for leaf disease classification. The datasets, the pretrained CNN models, and the proposed model are explained in detail in the following subsections.

## DATASET

PlantVillage (*Mohanty, 2016*) is a standard benchmark dataset used in plant disease classification research. It contains high-quality, labeled images with over 50,000 images spanning multiple plant species and disease categories, providing sufficient data for robust model training and validation. This publicly available dataset contains images of 14 different plant species (apple, blueberry, cherry, corn, grape, orange, peach, bell pepper, potato, raspberry, soybean, pumpkin, strawberry, and tomato). The dataset covers 38 disease classes (17 fungal diseases, four bacterial diseases, two mold diseases, two virus-related diseases, one worm-related disease, and 12 healthy diseases), providing diversity within a standardized framework. In this study, grape, apple and potato leaf images from the PlantVillage dataset were used for disease classification. Grapes are an important source of income in terms of agricultural production. Apples are among the most consumed fruits worldwide. They are grown as important agricultural products in many countries. They can grow in different climates. Potatoes, staple foods around the world, can grow in different climates and soil conditions. They are a fundamental part of many people's diets.

The grape dataset consists of 4,062 leaf images of four classes: healthy, leaf blight, tinder disease (esca), and black rot. The Apple leaf dataset consists of four classes, healthy, black rot, apple rust, and apple scab, including 3,172 leaf images. The potato leaf dataset consists of 2,152 leaf images in three classes: late blight, early blight, and healthy. Figure 1 shows healthy and diseased leaf images from the grape, apple, and potato datasets. For the grape dataset, the distribution encompasses 1,180 images representing black rot, 1,383 esca

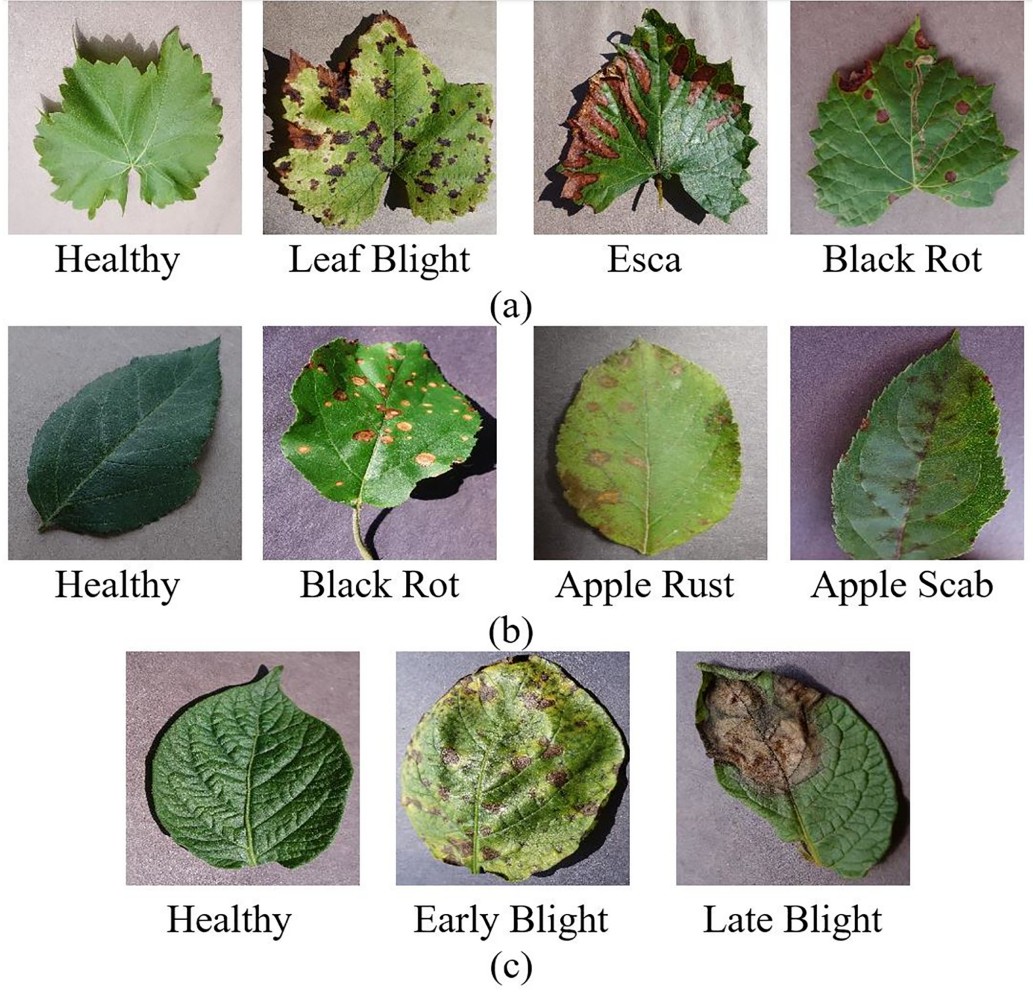

**Figure 1 Healthy and diseased (A) grape (B) apple (C) potato leaf images.** PlantVillage Dataset (*Mohanty, 2016*).

images, 1,076 images displaying leaf blight, and 423 images of healthy samples. The apple dataset contains 630 images of apple scab, 621 images of black rot, 276 images of cedar apple rust, and 1,645 images of healthy samples. The potato dataset comprises 1,000 images of early blight, 1,000 images of late blight, and 152 images of healthy samples. This distributional imbalance can cause the models to underfit classes with fewer samples and overfit them to the features of the majority class. For this reason, oversampling was performed with the SMOTE method to solve the data imbalance problem.

## METHOD

In this study, datasets from three plant species (grape, apple, potato) were utilized for the early detection of plant diseases. A flow diagram of the proposed plant disease classification method is given in Fig. 2. In the preprocessing step, the images were resized to 224 × 224 and normalized. In the data partitioning phase, the images are randomly divided into 80% training and 20% validation sets. This specific ratio was selected

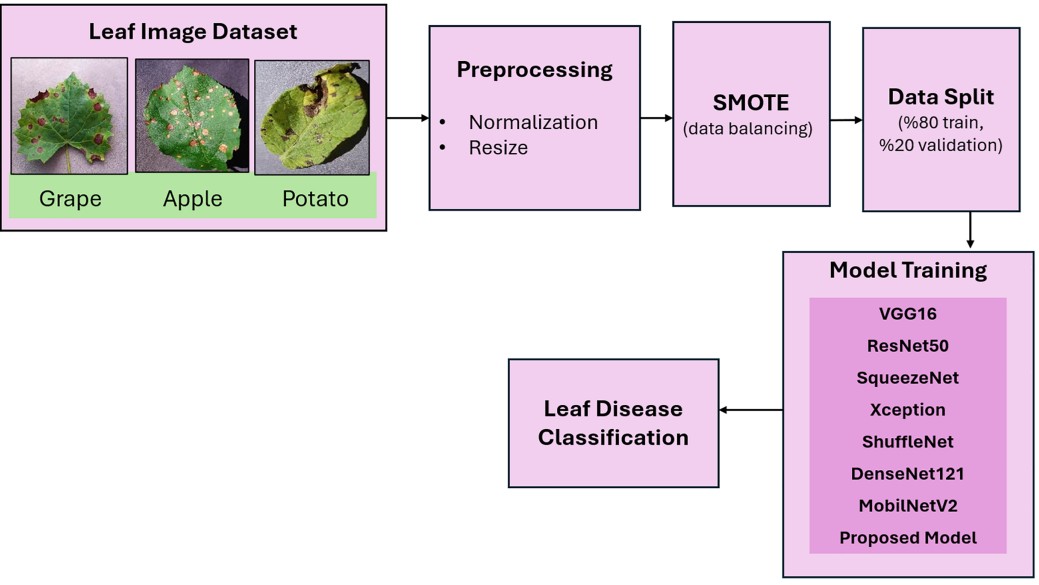

**Figure 2 Flow diagram of the proposed method for plant leaf disease classification.** Leaf images are taken from PlantVillage Dataset (*Mohanty, 2016*).

according to a comprehensive review of applications in the literature. The SMOTE method was applied to balance the distribution of data between classes. The pretrained CNN models- VGG16, ResNet50, SqueezeNet, Xception, ShuffleNet, DenseNet121, and MobileNetV2- were subsequently implemented for leaf disease classification. Finally, inspired by the success of the lightweight model SqueezeNet, a more lightweight model, GAPNet, was proposed for the classification of leaf diseases.

## Synthetic minority oversampling technique (SMOTE)

In training CNN models, it is very common to use data augmentation methods (GAN-based methods, ImageDataGenerator, or oversampling) for datasets. GAN-based solutions require significant amounts of data, are difficult to tune, and may suffer from model collapse (*Dablain, Krawczyk & Chawla, 2023*). ImageDataGenerator-based data augmentation does not have a positive effect on performance in all the cases. The effect of data augmentation varies depending on the size and characteristics of the dataset used (*Firnando et al., 2024*). On the other hand, data imbalance between classes may lead to a decrease in the performance of classification models. This imbalance causes the model to underfit the class with fewer samples and to overfit the features of the majority class.

In this study, a data preprocessing algorithm called SMOTE (*Elreedy & Atiya, 2019*) is used to balance the data distribution in the image dataset (*Pamungkas, Ramadani & Njoto, 2024*; *Özdemir, Polat & Alhudhaif, 2021*). SMOTE analyzes the nearest neighbors of instances belonging to the minority class to create new synthetic data on the basis of the relationships between these instances. The SMOTE method is commonly used because of its simplicity and high success rate in applications (*Elreedy & Atiya, 2019*). This method not only increases the number of instances belonging to the minority class but also adds

random variations to allow variability between instances (*Özden, 2023*). The synthetic data generation using the SMOTE method is given in Eq. (1). In the equation, $X_i'$ represents the newly synthesized instance, while $X_j$ is selected from the $k$ nearest neighbors of $X_i$. $\lambda$ is a random number in the range of [0–1] (*Elreedy & Atiya, 2019*).

$$X_i' = X_i + \lambda(X_j - X_i) \tag{1}$$

The SMOTE method balances classes by generating synthetic images based on the class with the most images. The original and SMOTE-generated leaf image examples for the grape, apple and potato datasets are shown in Fig. 3.

## Convolutional neural networks

CNNs are variants of multilayer perceptrons that are designed to emulate the behavior of the visual cortex. These methods can achieve successful results in image classification tasks because of their multilayered structures. CNNs generally consist of convolution layers, pooling layers, and fully connected layers. In the convolution layers, several filtering operations are performed to extract features from the input image. The process of extracting distinct features is performed by sliding the filter over the image. The pooling layers perform pooling operations to reduce the dimensionality of the data. In the fully connected layer, matrix-formatted data are flattened into a vector to perform the learning process.

In this study, pretrained CNN models, VGG16, ResNet50, SqueezeNet, Xception, ShuffleNet, DenseNet121, and MobileNetV2, were utilized. Later, a lightweight and fast CNN model, GAPNet, was proposed for grape, apple and potato leaf disease classification.

## VGG16

VGG16 was proposed by *Simonyan & Zisserman (2014)* from the University of Oxford in 2014. This network was trained on the ImageNet database. The VGG16 network consists of 13 convolution layers and three fully connected layers. The model includes five convolutional blocks. While the first and second blocks consist of two convolution layers, the last three blocks consist of three convolution layers. In each block, the pooling operation is applied after the convolution layers. Then, classification is performed through the fully connected layers.

## ResNet50

The ResNet architecture aims to address the vanishing gradient problem during the training process of CNNs, which leads to a decrease in the accuracy of the models (*Mukti, 2019*). To solve this problem, residual connections are used. Residual connections are a type of skip-connection that learns residual functions with reference to the layer inputs instead of learning unreferenced functions (*He et al., 2016*). The use of residual connections allows for learning with deeper networks.

ResNet50 is a residual network architecture consisting of 50 layers, including convolutional layers, identity blocks, convolutional blocks, and fully connected layers. The

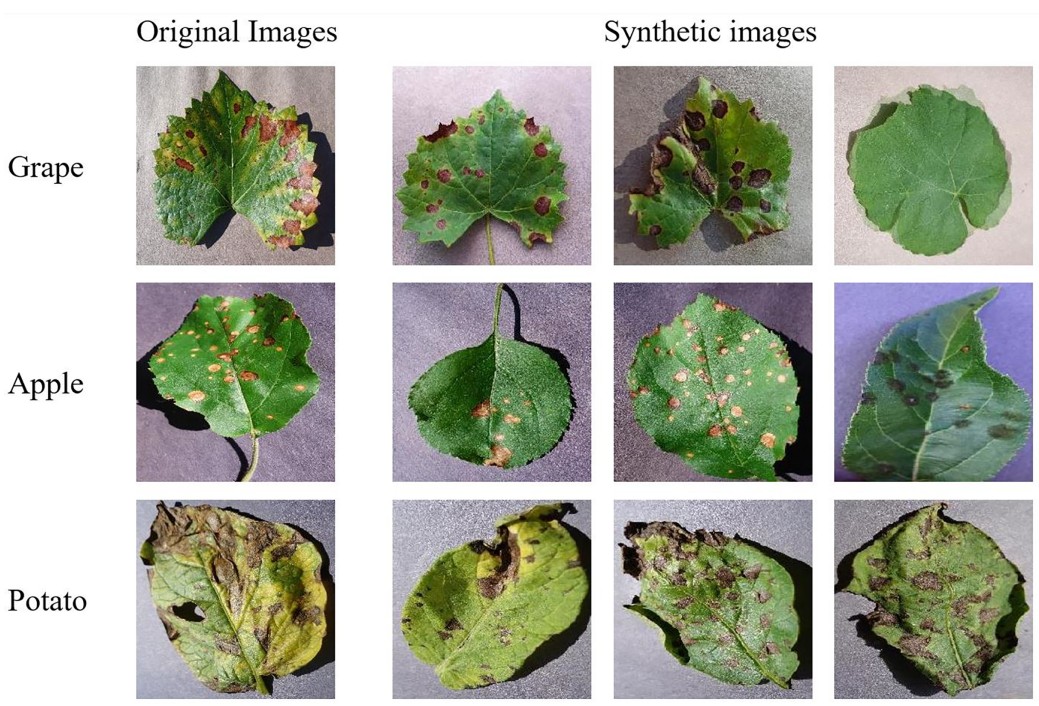

**Figure 3 Original and SMOTE-generated leaf image samples.** PlantVillage Dataset (*Mohanty, 2016*).

convolutional layers extract features from the input image. These features are processed and transformed using identity and convolutional blocks. Finally, the fully connected layers are utilized for classification.

## SqueezeNet

SqueezeNet is an 18-layer network architecture with a small number of parameters. The model was created with three design strategies: filter reduction by replacing the 3 × 3 filter with a 1 × 1 filter, reducing the input channels, and subsampling at the end of the network to reduce the size (*Hidayatuloh, 2018*). SqueezeNet uses the fire module, which combines 1 × 1 and 3 × 3 filters to reduce the number of parameters (*Iandola, 2016*). SqueezeNet starts with a convolution layer, followed by eight fire modules, and ends with a final convolution layer.

## Xception

The Xception model is constructed with 36 convolutional layers. The convolutional layers are separated from each other by residual connections (*Chollet, 2017*). The model demonstrates deeply separable convolution features, which independently perform point-to-point convolution in each channel of the input data (*Chen, 2021*). In the Xception architecture, data pass through an entry flow, followed by a middle flow, and finally an exit flow.

## ShuffleNet

ShuffleNet was proposed by MEGVII researchers in 2017. It was designed to create a low-cost lightweight model. In ShuffleNet, the channel mixing method, which allows feature maps belonging to different channel groups to exchange information without increasing the computational load, is used (*Chen, 2022*).

The ShuffleNetV2 architecture provides a lower cost and higher accuracy. This approach can achieve effective results by reducing the number of model parameters and calculations. The concept of group convolution, which divides the network, input, and output channels into more than one group and performs convolution operations within each group, was used (*Zhou, 2024*). This block consists of a combination of pointwise group convolution, a channel shuffling operation, and depthwise convolution. The channel shuffling operation enables information exchange across groups, enhancing the representation capacity of the network (*Ma, 2018*).

## DeneseNet121

The DenseNet architecture uses a block structure where layers are densely connected in a feedforward manner, and it employs bottleneck layers to reduce the number of parameters. There are different DenseNet architectures, such as DenseNet121, DenseNet160, and DenseNet201. In this work, the DenseNet121 model (*Huang et al., 2017*) was used because it has fewer parameters than the other models. This model consists of 121 layers, including 120 convolutional layers and one fully connected layer. The DenseNet121 architecture consists of four dense blocks and three transition layers.

## MobileNetV2

MobileNetV2 was proposed by Google researchers in 2018. The MobileNet architecture is based on depthwise separable convolution. The MobileNetV2 model was developed by adding inverted residuals with linear bottleneck modules to the MobileNetV1 model (*Indraswari, 2022*). This model is constructed from convolutional, inverted residual bottleneck, and pointwise convolution layers.

The details of the model architecture designed for plant disease classification using these seven pretrained networks are given in Table 1. Initially, 2D convolution is applied to the input image. Then, feature extraction is performed using a pretrained model. Finally, the classification phase consists of dropout and dense layers, as shown in Table 1.

## Proposed model–GAPNet

The SqueezeNet model demonstrates high accuracy despite having fewer parameters and complexity among state-of-the-art networks. The significantly smaller memory requirements of SqueezeNet make it more suitable for deployment with limited computing resources. Furthermore, the design of SqueezeNet allows easier modification and integration of fire components than more complex architectures. It allows for improvements without extensive architectural redesign. Among the seven trained models, SqueezeNet showed comparable performance despite the small number of parameters. For these reasons, SqueezeNet was used as the basis for the proposed GAPNet model.

| Table 1 Details of the plant leaf disease classification model. |
| --- |
| **Model architecture** |
| Input_layer (224 × 224 × 3) |
| Conv2D (Convolution Layer) |
| Pretrained models |
|   (VGG16, ResNet50, SqueezeNet, Xception, ShuffleNet, DenseNet121, MobilNetV2) |
| Flatten (Flatten Layer) |
| Dropout |
| Dense layer (Activation = 'Relu') |
| Batch normalization |
| Dropout |
| Output layer (Dense (Activation = 'Softmax')) |

In this study, a lightweight and fast CNN model, GAPNet, was proposed for disease classification from leaf images. This network was developed as a simplified version of the SqueezeNet model. Figure 4 shows the proposed GAPNet architecture. In the proposed model, convolution and max pooling were applied to the input images. It is followed by two fire modules, one max pooling module, two fire modules, one max pooling module, and two fire modules, as shown in Fig. 4. There are eight Fire modules in the original SqueezeNet network. In GAPNet, six of the eight Fire modules were preserved, and the others were removed. Therefore, the number of parameters is reduced from 724,560 to only 337,872. The fire modules are followed by the max pooling and dropout layers. After the last convolution layer, global average pooling and classification were applied.

The number of Fire modules in the proposed architecture was determined experimentally. In this process, the fire modules were removed one by one, and the contribution of each module to the overall system was evaluated. Therefore, the eight fire modules in the original SqueezeNet model were reduced to seven, six, five, four, three, and two, and the effect of each fire module on model performance was examined. The results showed that six fire modules gave optimum results in terms of the success rate and number of parameters.

In this model, fire modules enable the design of a lightweight network. In the fire module, some of the 3 × 3 convolutions are replaced with 1 × 1 convolutions to reduce the number of parameters. This module consists of squeeze layers (1 × 1 convolutions) and expand layers (1 × 1 and 3 × 3 convolutions). The proposed lightweight CNN model can achieve comparable results to those of other high-performing CNN models.

The number of parameters and the sizes of the CNN models used in the study are given in Table 2. As shown in the table, SqueezeNet has a very low number of parameters compared with the other pretrained networks. This shows that this network is applicable in systems with limited resources. On the other hand, the proposed GAPNet has 58.88% fewer parameters than the original model. GAPNet has the smallest storage requirement of 1.29 MB, which is much lighter than those of the other models. SqueezeNet has the second smallest size of 2.76 MB, whereas ResNet50 and Xception have the largest storage

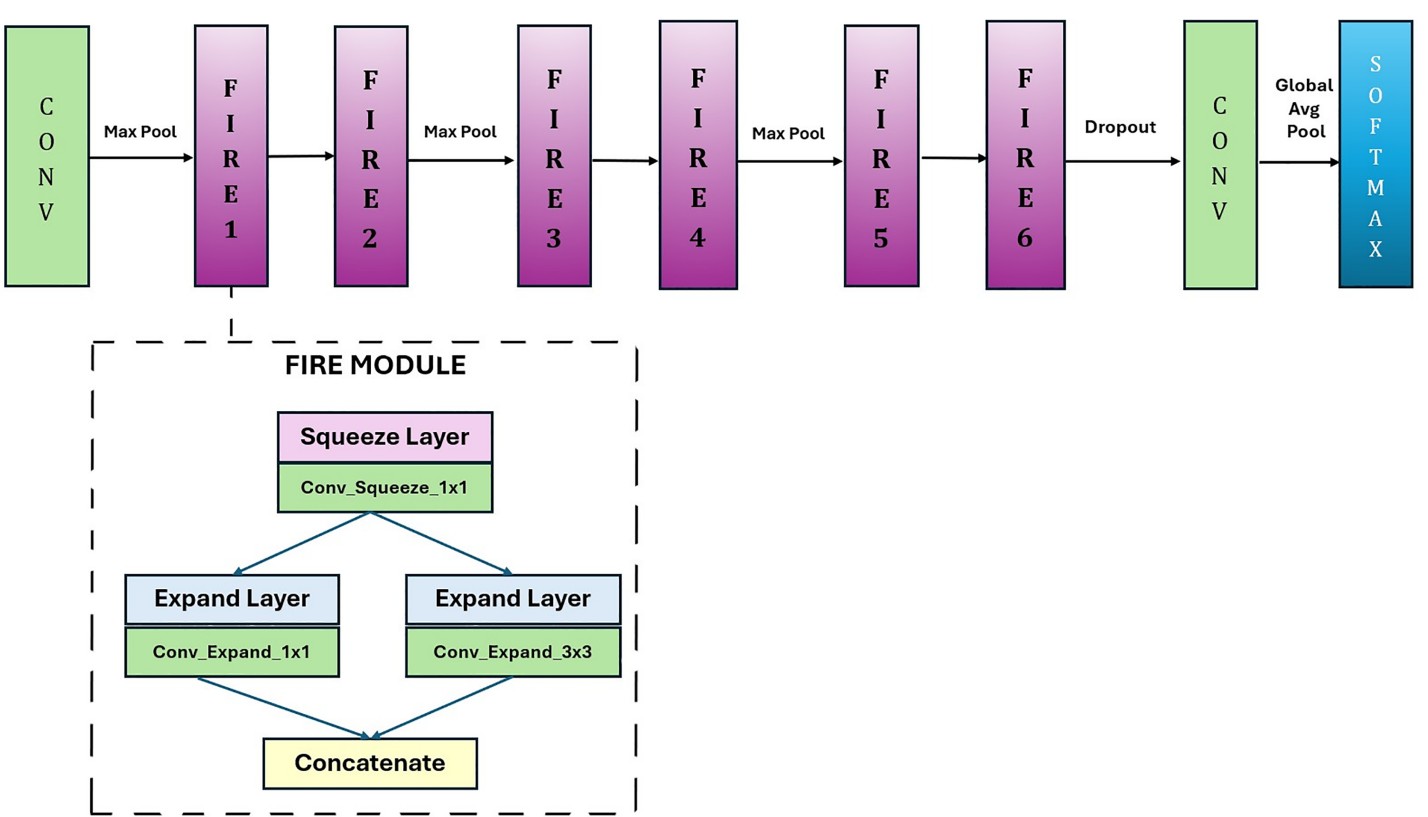

**Figure 4 GAPNet architecture.**

**Table 2 Number of parameters and size of models.**

| Model | Input size | Number of parameters | Model size |
|---|---|---|---|
| VGG16 | 224 × 224 | 14.815.128 | 80.64 MB |
| ResNet50 | 224 × 224 | 49.280.216 | 188.00 MB |
| SqueezeNet | 224 × 224 | 724.560 | 2.76 MB |
| Xception | 224 × 224 | 46.553.984 | 177.59 MB |
| ShuffleNet | 224 × 224 | 4.022.840 | 15.35 MB |
| DenseNet121 | 224 × 224 | 7.306.136 | 27.87 MB |
| MobileNetV2 | 224 × 224 | 18.316.696 | 69.87 MB |
| Proposed model | 224 × 224 | 337.872 | 1.29 MB |

requirements of 188.00 and 177.59 MB, respectively. However, the proposed GAPNet was found to be very successful in classifying grape, apple, and potato leaf diseases.

## Experimental results

In this study, disease classification was performed on grape, apple and potato leaf images with pretrained VGG16, ResNet50, SqueezeNet, Xception, ShuffleNet, DenseNet121, MobileNetV2 and the proposed GAPNet networks. The data imbalance problem was

addressed using the SMOTE method. The images are then randomly divided into 80% training and 20% validation sets.

In this study, experiments were conducted in the Google Colab environment with an A100 GPU. The hyperparameters used for training the CNN models are given in Table 3. Accuracy, precision, F1-score, and recall metrics were used to evaluate the performance of the models. The confusion matrix given in Fig. 5 is used to calculate these metrics. In the figure TP is the number of positive samples identified as positive by the classifier and TN is the number of negative samples identified as negative by the classifier. The number of examples whose true label is negative but predicted as positive by the classifier is expressed as FP, and the number of examples whose true label is positive but predicted as negative by the classifier is expressed as FN. The mathematical expressions for accuracy, F1-score, precision, and recall are given in Eqs. (2)–(5).

$$Accuracy = \frac{TP + TN}{TP + TN + FP + FN} \tag{2}$$

$$Precision = \frac{TP}{TP + FP} \tag{3}$$

$$Recall = \frac{TP}{TP + FN} \tag{4}$$

$$F1\text{-}score = \frac{2 * (Precision * Recall)}{(Precision + Recall)}. \tag{5}$$

The classification performances of the VGG16, ResNet50, SqueezeNet, Xception, ShuffleNet, DenseNet121, and MobileNetV2 networks on the grape, apple and potato leaf disease datasets are shown in Table 4. For grape leaf diseases, MobileNetV2 and DenseNet121 have higher classification accuracies than the other models. The highest classification performance for grape leaf disease images is achieved with an accuracy of 100% and an F1-score of 100% with these models. SqueezeNet, which has significantly fewer parameters than the other models, provides promising results, with an accuracy of 99.55% and an F1-score of 99.55% on the grape dataset.

For the Apple dataset, VGG16, ResNet50, MobileNetV2, and DenseNet121 achieved higher accuracies than the other models. For apple leaf images, the highest disease classification performance is achieved with an accuracy of 100% and an F1-score of 100% when the VGG16, ResNet50, MobileNetV2, and DenseNet121 networks are used. The results in Table 4 show that SqueezeNet also provides comparable results, with an accuracy of 99.31% and an F1-score of 99.31%.

The classification results of the networks on the Potato dataset indicate that VGG16, ResNet50, DenseNet121, and Xception have higher accuracies than the other models. The highest disease classification performance for potato leaf images was achieved with an accuracy of 100% and an F1-score of 100% via the VGG16, ResNet50, DenseNet121, and Xception models. Similarly, the lightweight SqueezeNet model achieved significant disease classification performance, with an accuracy of 99.86% and an F1-score of 99.83%.

After the single-plant leaf disease classification task was evaluated, samples from three species were combined, and multiplant leaf disease classification was performed via

**Table 3 Parameters for training models.**

| Parameter name | Value |
|---|---|
| Input size | $224 \times 224 \times 3$ |
| Batch size | 64 |
| Epoch number | 100 |
| Optimizer | Adam |
| Learning rate | 0.0001 |
| Loss function | Categorical cross entropy |
| Dropout rate | 0.5 |

|  | Positive | Negative |
|---|---|---|
| Positive | True Positive (TP) | False Positive (FP) |
| Negative | False Negative (FN) | True Negative (TN) |

Predicted Label

True Label

**Figure 5  Confusion matrix.**     

pretrained models. The results in Table 4 show that the DenseNet121 and Xception networks are more successful in 11-class classification.

Table 4 presents a comprehensive comparison of performance metrics (accuracy, precision, recall, and F1-score) for various CNN architectures across grape, apple, potato, and combined datasets under both original and SMOTE-balanced class distributions. The experimental results demonstrate that addressing class imbalance through oversampling yielded consistent performance improvements across all the evaluation metrics. Notably, while improvements in accuracy are observed, the precision, recall, and F1-score metrics exhibited more substantial gains after the implementation of the SMOTE method. These results clearly indicate that addressing class imbalance contributes to model performance in leaf disease classification tasks. The results of the experiments show that the lightweight SqueezeNet model yields comparable results in both single and multiplant leaf disease classification.

In this study, the number of Fire modules in the proposed GAPNet model was determined experimentally. The original SqueezeNet model consists of eight fire modules. In this study, the number of fire modules was reduced, the plant leaf disease classification performance was evaluated, and the results are given in Table 5. All the models were trained for 150 epochs, and oversampling was applied to the training samples. A total of six Fire modules gave optimum results in terms of success and number of

**Table 4 Classification performances of pre-trained CNN models on single and multi plant leaf disease.**

| | | VGG16 | SqueezeNet | ResNet 50 | MobileNet V2 | DenseNet 121 | Xception | ShuffleNet | SMOTE |
|---|---|---|---|---|---|---|---|---|---|
| **Grape** | Accuracy | 99.9 | 99.55 | 99.9 | 100 | 100 | 99.45 | 99 | + |
| | | 100 | 99.75 | 99.87 | 100 | 100 | 98.4 | 98.76 | − |
| | F1-score | 99.91 | 99.55 | 99.91 | 100 | 100 | 99.47 | 99.02 | + |
| | | 99.7 | 99.04 | 99.9 | 99.21 | 99.9 | 98.31 | 98.89 | − |
| | Precision | 99.91 | 99.55 | 99.91 | 100 | 100 | 99.47 | 99.02 | + |
| | | 99.73 | 99.05 | 99.89 | 99.18 | 99.89 | 98.28 | 98.75 | − |
| | Recall | 99.91 | 99.56 | 99.91 | 100 | 100 | 99.47 | 99.03 | + |
| | | 99.68 | 99.06 | 99.9 | 99.72 | 99.9 | 98.34 | 99.04 | − |
| **Apple** | Accuracy | 100 | 99.31 | 100 | 100 | 100 | 99.84 | 98.63 | + |
| | | 100 | 98.74 | 100 | 98.43 | 100 | 99.84 | 97.48 | − |
| | F1-score | 100 | 99.31 | 100 | 100 | 100 | 99.84 | 98.63 | + |
| | | 99.86 | 97.53 | 100 | 96.22 | 99.45 | 99.86 | 96.85 | − |
| | Precision | 100 | 99.31 | 100 | 100 | 100 | 99.84 | 98.63 | + |
| | | 99.92 | 97.01 | 100 | 96.38 | 99.69 | 99.92 | 96.96 | − |
| | Recall | 100 | 99.31 | 100 | 100 | 100 | 99.84 | 98.63 | + |
| | | 99.81 | 98.10 | 100 | 96.30 | 99.22 | 99.81 | 96.76 | − |
| **Potato** | Accuracy | 100 | 99.86 | 100 | 99.00 | 100 | 100 | 98.66 | + |
| | | 100 | 99.77 | 99.77 | 99.76 | 100 | 99.77 | 98.61 | − |
| | F1-score | 100 | 99.83 | 100 | 99.01 | 100 | 100 | 98.69 | + |
| | | 100 | 97.48 | 99.14 | 99.83 | 100 | 99.33 | 96.89 | − |
| | Precision | 100 | 99.83 | 100 | 99.02 | 100 | 100 | 98.71 | + |
| | | 100 | 97.53 | 99.66 | 99.83 | 100 | 98.85 | 95.61 | − |
| | Recall | 100 | 99.84 | 100 | 99.03 | 100 | 100 | 98.66 | + |
| | | 100 | 97.48 | 98.64 | 99.83 | 100 | 99.83 | 98.34 | − |
| **Grape + Apple + Potato** | Accuracy | 99.97 | 99.28 | 99.97 | 99.97 | 100 | 100 | 99.80 | + |
| | | 99.78 | 98.8 | 99.94 | 99.84 | 100 | 99.94 | 99.46 | − |
| | F1-score | 99.97 | 99.28 | 99.97 | 99.97 | 100 | 100 | 99.80 | + |
| | | 99.40 | 97.84 | 99.96 | 99.73 | 100 | 99.83 | 99.39 | − |
| | Precision | 99.97 | 99.29 | 99.97 | 99.97 | 100 | 100 | 99.80 | + |
| | | 99.82 | 97.71 | 99.96 | 99.62 | 100 | 99.97 | 99.55 | − |
| | Recall | 99.97 | 99.27 | 99.97 | 99.97 | 100 | 100 | 99.80 | + |
| | | 99.04 | 98.01 | 99.96 | 99.84 | 100 | 99.69 | 99.24 | − |

parameters. Therefore, in this study, the GAPNet model was designed to consist of six Fire modules.

The performance results of the proposed GAPNet model for the grape, apple, potato and grape + apple + potato datasets are given in Table 6. Compared with state-of-the-art CNN models, GAPNet, a lighter model, has provided promising single and multiplant leaf disease classification results. The proposed model showed the highest performance, with an accuracy of 99.72% and an F1-score of 99.73% for the 4-class grape dataset. GAPNet achieved accuracies of 99.53% and 99.83% for apple and potato leaf disease classification, respectively. For the 11-class dataset, 99.64% classification accuracy was achieved.

**Table 5 Classification accuracies according to the number of fire modules.**

| Number of fire modules | Number of parameters | Dataset | | | |
|---|---|---|---|---|---|
| | | Grape | Apple | Potato | G + A + P |
| 2 | 26.160 | 95.84 | 97.59 | 98.83 | 96.57 |
| 3 | 72.016 | 98.28 | 98.36 | 99.00 | 98.06 |
| 4 | 121.456 | 99.55 | 98.86 | 99.50 | 99.14 |
| 5 | 226.848 | 99.54 | 98.91 | 99.83 | 99.25 |
| 6 | 337.872 | 99.72 | 99.53 | 99.83 | 99.64 |
| 7 | 530.967 | 99.81 | 99.53 | 99.83 | 99.69 |
| 8 | 724.560 | 99.63 | 99.84 | 100 | 99.61 |

**Table 6 Proposed model performance results on all datasets.**

| Plant | Model | Accuracy | F1-score | Precision | Recall |
|---|---|---|---|---|---|
| Grape | Proposed model | 99.72 | 99.73 | 99.73 | 99.73 |
| Apple | Proposed model | 99.53 | 99.54 | 99.54 | 99.53 |
| Potato | Proposed model | 99.83 | 99.83 | 99.83 | 99.84 |
| Grape + Apple + Potato | Proposed model | 99.64 | 99.64 | 99.64 | 99.64 |

The confusion matrices for the validation sets of the grape, apple, potato and grape + apple + potato datasets obtained using the GAPNet model are shown in Fig. 6. Additionally, the classification performance of SqueezeNet and the proposed model for all classes is given in detail in Table 7. The 'esca' class in the grape dataset, the 'apple rust' class in the apple dataset, and the 'early blight' and 'late blight' classes in the potato dataset are classified with 100% accuracy using SqueezeNet. In addition, the model has lower performance in the 'eaf blight' and 'apple scap' classes when compared with other classes. For the GAPNet model, the 'healthy', 'esca' and 'black rot' classes in the grape dataset, the 'black rot' and 'apple rust' classes in the apple dataset, and the 'late blight' and 'healthy' classes in the potato dataset are classified with 100% accuracy.

The accuracy and loss graphics of the training and validation sets with the proposed GAPNet model on the grape, apple, potato and grape + apple + potato datasets are shown in Fig. 7. The experimental results indicate that the GAPNet model performs successfully on all the datasets. There was no overfitting, and fluctuations in the accuracy and loss values decreased after 50 epochs.

The performance comparison of SqueezeNet and the proposed GAPNet models on the grape, apple, potato and grape + apple + potato datasets is shown in Fig. 8. The proposed lightweight GAPNet model, with 337.872 parameters, achieves classification performance comparable with that of the SqueezeNet model. SqueezeNet has a very low number of parameters compared with the other pretrained networks (Table 2). This shows that this network is applicable in systems with limited resources. On the other hand, the proposed GAPNet has 58.88% fewer parameters than the original model. However, this lightweight

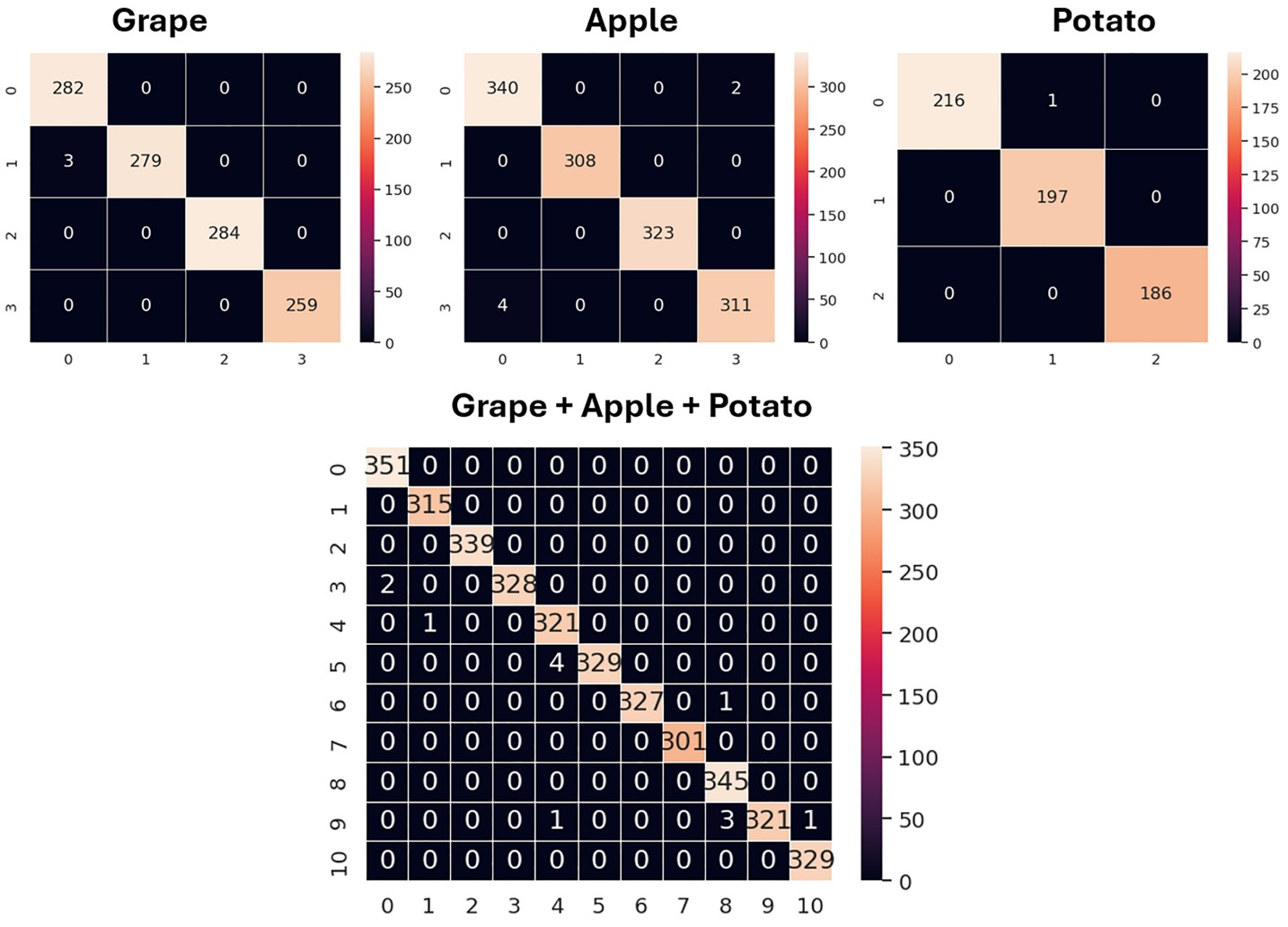

**Figure 6** Confusion matrices obtained with GAPNet on apple, potato, and grape datasets (validation).

network was found to be very successful in single and multiplant leaf disease classification for grape, apple and potato plants, as shown in Fig. 8.

Table 8 shows the computational efficiency measurements for GAPNet compared with the seven baseline models. To ensure fair comparison, the tests were run on the same hardware. GAPNet demonstrates high computational efficiency with the fastest training time of 6 s and 26 ms per epoch, which is 14% faster than that of SqueezeNet (7 s, 30 ms) and 45% faster than that of ShuffleNet (11 s, 47 ms). GAPNet is also significantly more efficient than the other models. Furthermore, GAPNet achieves the most efficient inference speed, with a test time of 0.00083 ms per sample. This is 40% faster than SqueezeNet (0.00138 ms), 50% faster than VGG16 (0.00166 ms), 67% faster than ShuffleNet (0.00249 ms), 70% faster than both ResNet50 and Xception (0.00277 ms), 63% faster than MobileNetV2 (0.00222 ms), and 88% faster than DenseNet121 (0.00693 ms).

**Table 7** SqueezeNet and GAPNet performance evaluation on apple, potato and grape datasets (validation).

| Plant | Class name | Accuracy (%) | |
|---|---|---|---|
| | | **SqueezeNet** | **GAPNet** |
| Grape | Healthy | 99.66 | 100 |
| | Leaf blight | 98.94 | 98.93 |
| | Esca | 100 | 100 |
| | Black rot | 99.61 | 100 |
| Apple | Apple scab | 98.51 | 99.41 |
| | Black rot | 99.67 | 100 |
| | Apple rust | 100 | 100 |
| | Healthy | 99.08 | 98.41 |
| Potato | Early blight | 100 | 99.53 |
| | Healthy | 99.49 | 100 |
| | Late blight | 100 | 100 |

These significant performance improvements make GAPNet particularly suitable for deployment in resource-constrained agricultural settings and real-time processing where computational efficiency is crucial.

A comparison of the classification accuracy of the proposed GAPNet model with the literature is shown in Table 9. Compared with the methods in the literature, the proposed GAPNet model is quite successful in classifying potato leaf diseases. Additionally, it showed comparable performance to those of previous studies in classifying grape and apple leaf diseases. As shown in the comparison table, studies in the literature generally develop models for single plant leaf disease classification. The number of studies conducted for multiplant leaf disease classification is quite low. From this perspective, the proposed model successfully classifies leaf diseases for single and multiplant species.

The performance of the proposed GAPNet model is also evaluated on the New Plant Diseases (*Bhattarai, 2019*) and Plant Pathology 2020 (*Kaeser-Chen, 2020*) datasets. New Plant Diseases is a publicly available dataset created by applying data augmentation to the original PlantVillage dataset. Plant Pathology (*Kaeser-Chen, 2020*) consists of apple leaf disease images obtained under real-world conditions with complex backgrounds. The images are captured under various angles, illuminations, surfaces, and noise conditions. The dataset contains four classes: apple scab, cedar apple rust, multidisease, and healthy.

The proposed GAPNet model showed superior performance, with 99.77% validation accuracy on the New Plant Disease dataset (on grape + apple + potato leaf images). These results prove that the model shows high success in images obtained under controlled environments such as ideal lighting conditions and homogeneous backgrounds.

In addition, a predictable drop in model performance was observed on the Plant Pathology dataset. A validation accuracy of 92.78% was achieved for the 3-class configuration, and 87.95% was achieved for the 4-class configuration on these complex

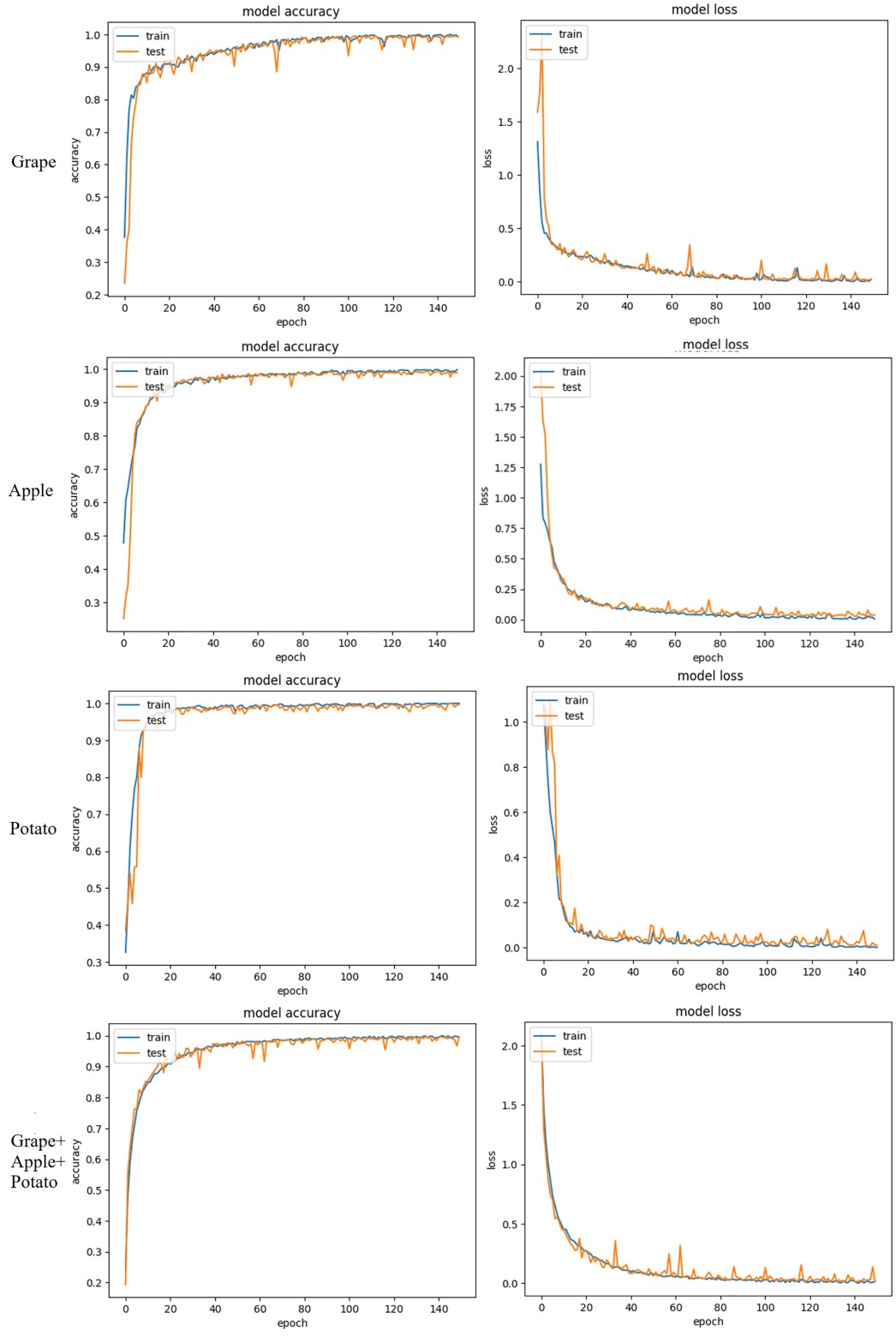

**Figure 7 Accuracy and loss graphs of GAPNet on apple, potato, and grape datasets.**

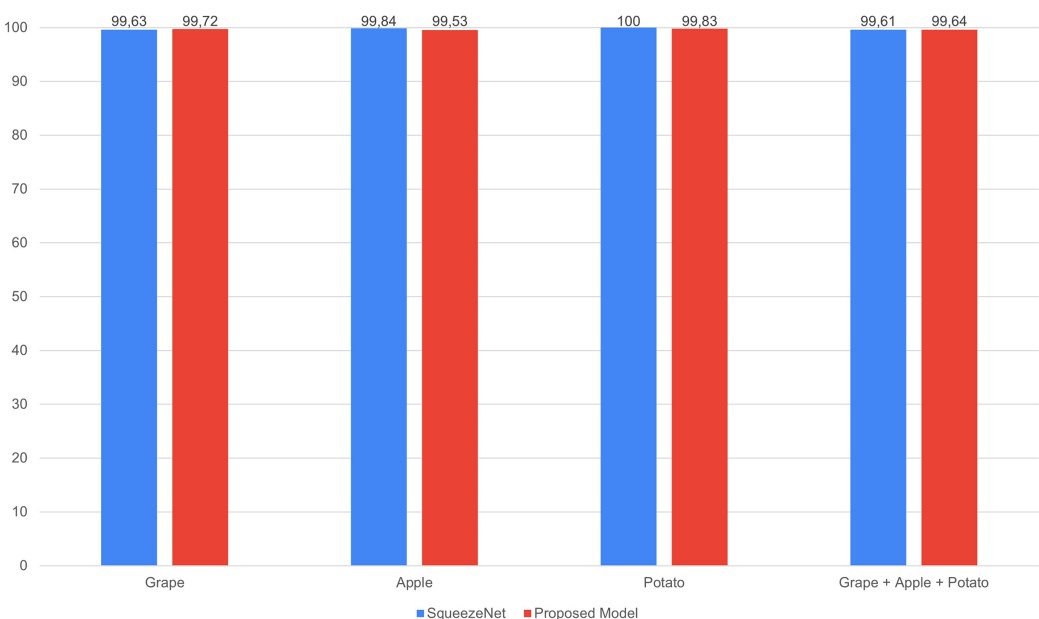

Figure 8 Performance comparison of SqueezeNet and GAPNet.

Table 8 Training and validation times of the model.

|  | Model | Duration of training (per epoch) | Duration of validation (per sample) |
|---|---|---|---|
| G + A + P | MobilNetV2 | 13 s 58 ms | 0.00222 ms |
|  | DenseNet121 | 27 s 119 ms | 0.00693 ms |
|  | ResNet50 | 21 s 94 ms | 0.00277 ms |
|  | VGG16 | 27 s 117 ms | 0.00166 ms |
|  | Xception | 27 s 118 ms | 0.00277 ms |
|  | SqueezeNet | 7 s 30 ms | 0.00138 ms |
|  | ShuffleNet | 11 s 47 ms | 0.00249 ms |
|  | GAPNet | 6 s 26 ms | 0.00083 ms |

images. These results reflect the difficulties of image classification in real-world conditions. However, compared with similar studies in the literature, the proposed method produces competitive results. The GAPNet model achieved a validation accuracy of 92.78%, significantly improving the performance of 87.31% achieved by *Fenu & Malloci (2021)* on the Plant Pathology dataset (three classes). Owing to its low number of parameters, the GAPNet model performed well, with 87.95% validation accuracy on the 4-class Plant Pathology dataset, which is very close to the 88.70% accuracy reported in the study of *Sapna et al. (2023)*. These results show that GAPNet is competitive with methods in real-world plant disease classification with complex background literature.

**Table 9 Literature comparison of GAPNet model.**

| Method | Dataset | Grape | Apple | Potato | G + A + P |
|---|---|---|---|---|---|
| *Nagaraju, Swetha & Stalin (2020)* | Data collected from google | 97.87% | 97.87% | – | – |
| *Çetiner (2021)* | Researchers' data | – | 99.00% | – | – |
| *Reddy & Neeraja (2022)* | Plant village | 96.75% | 96.89% | 97.20% | – |
| *Nagi & Tripathy (2022)* | Plant village | 98.40% | – | – | – |
| *Pradhan (2022)* | Plant village | – | 98.75% | – | – |
| *Babalola, Kpai & Toygar (2023)* | Plant village | – | 99.56% | – | – |
| *Sood & Singh (2024)* | Plant village | 99.88% | – | – | – |
| *Sofuoğlu & Bırant (2024)* | Plant village | – | – | 98.28% | – |
| *Upadhyay & Gupta (2024)* | Plant village | – | 98.94% | – | – |
| *Sarfarazi, Zefrehi & Toygar (2024)* | Plant village | – | – | 98.61% | – |
| | Potato leaf dataset | | | 97.78% | |
| *Banjar et al. (2025)* | Plant village | – | 99.00% | – | – |
| Proposed method | Plant village | 99.72% | 99.53% | 99.83% | 99.64% |

## CONCLUSION

In this study, GAPNet, a lightweight and effective CNN model for the classification of leaf diseases, is proposed. GAPNet was developed as a smaller version of the SqueezeNet architecture. In the study, seven different pretrained CNN models, VGG16, ResNet50, SqueezeNet, Xception, ShuffleNet, DenseNet121, and MobileNetV2, which are widely used for the classification of leaf diseases, were evaluated. After this comparative analysis, the GAPNet model, a simplified SqueezeNet architecture specifically designed for the classification of leaf diseases, is proposed. GAPNet is designed to have only 337.872 parameters and has a lightweight and fast structure that is both computationally efficient and can operate with low resource requirements. This allows the model to be used on mobile devices and systems with limited hardware resources.

In this study, datasets consisting of leaf images collected from three different plant species (grape, apple, and potato) were used. Data oversampling was performed using the SMOTE method to eliminate the imbalance problem between the classes in the datasets. The experimental results revealed that the proposed GAPNet model can classify grape, apple, and potato leaf diseases with high accuracy. When existing studies in the literature are examined, models are generally developed by focusing on a single plant species. Studies that can classify leaf diseases of more than one plant species simultaneously are quite limited. In this context, the proposed GAPNet model has attracted attention because of its successful multiplant leaf disease classification performance and its lightweight structure. Consequently, the model has the potential to be used effectively in practical agricultural applications.

Although our study provides successful results for single and multiplant leaf disease classification, there are several limitations. The proposed model was trained on a dataset consisting of leaf images obtained in a laboratory environment. Consistent imaging conditions in PlantVillage allow us to focus on algorithmic improvements. The proposed

model is designed to detect diseases in three specific plant species. The model needs to be further developed to accurately and reliably detect leaf diseases in other plant species. However, when experiments were performed on leaf images taken in the wild, the model performance decreased. To overcome this limitation and use the model more effectively in real-world problems, a larger and more diverse dataset needs to be included in the experiments. Although our current model shows strong performance in disease detection for three plant species, we need to address its limited scope by implementing improvements that will enable successful leaf disease detection in a variety of plants.

Future work will aim to address these limitations. We aim to develop the proposed model to be compatible not only with three plant species but also with various plant species and complex backgrounds to evaluate its generalizability. We also aim to incorporate ensemble techniques to increase the generalizability of the model to different datasets.

## ACKNOWLEDGEMENTS

Claude.ai tool was used in the revision process of the manuscript.

### Funding
The authors received no funding for this work.

### Competing Interests
The authors declare that they have no competing interests.

### Author Contributions
- Özge Nur Özaras performed the experiments, performed the computation work, prepared figures and/or tables, authored or reviewed drafts of the article, and approved the final draft.
- Asuman Günay Yılmaz conceived and designed the experiments, analyzed the data, authored or reviewed drafts of the article, and approved the final draft.

### Data Availability
The data and code are available at GitHub and Zenodo:

- https://github.com/ozgenurr/GAPNet.git.

- Ozge Ozaras. (2025). ozgenurr/GAPNet: GAPNET (GAPNET). Zenodo. https://doi.org/10.5281/zenodo.15163686.

The datasets are publicly available at:

- Plant Village Dataset: https://github.com/spMohanty/PlantVillage-Dataset/tree/master/raw/color.

- New Plant disease Dataset: https://www.kaggle.com/datasets/vipooool/new-plant-diseases-dataset/data.

- Plant Pathology: https://www.kaggle.com/competitions/plant-pathology-2020-fgvc7/data.
## Supplemental Information

Supplemental information for this article can be found online at http://dx.doi.org/10.7717/peerj-cs.2941#supplemental-information.

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
