# Peer review of "GAPNet: Single and multiplant leaf disease classification method based on simplified SqueezeNet for grape, apple and potato plants"

_PeerJ Computer Science, doi:10.7717/peerj-cs.2941_

## Round 0.1 · original submission · Major Revisions

Based on the referee reports, I recommend a major manuscript revision. The author should improve the manuscript, carefully consider the reviewers' comments in the reports, and resubmit the paper.

Reviewer 1 ·

Basic reporting

See additional comment

Experimental design

See additional comment

Validity of the findings

See additional comment

Additional comments

1. In the Introduction, the contribution should be explained more clearly, the development of the model or focus on plant diseases as the object. If on the object, testing on the dataset should be richer, not limited to apple, grape and potato diseases. Several other datasets such as rice diseases, or other plants can be added.

2. The role of augmentation and GAPNet needs to be studied more deeply, which is more dominant with ablation studies.

3. SMOTE is not commonly applied to image augmentation, it is more appropriate to call it oversampling because it is generally on tabular data. Then it needs to be explained whether augmentation/oversampling is done on the CNN extraction feature results, if so, the paper https://doi.org/10.62411/jcta.11530 can be used as a reference. But if SMOTE is applied directly to the image, there must be a scientific justification why augmentation is not done with ImageDataGenerator, Albumentations or GAN.

4. Explain why the CNN basis was chosen, not Vision Transformers. The paper http://dx.doi.org/10.62411/jcta.10459 can be added as a reference

5. Give reasons why only the PlantVillage dataset is used.

6. Add a parameter configuration table for each model compared.

7. Explain in more detail the hypothesis of choosing SqueezeNet as the basis for GAPNet, not other models

8. The analysis of the results needs to be more in-depth, what is the relationship between dataset distribution and accuracy, precision, recall, f1, and specificity, and which measurement tool is more emphasized? See and cite the paper https://doi.org/10.62411/faith.2024-4


9. The conclusion section must be able to answer all research objectives,

Reviewer 2 ·

Basic reporting

*The manuscript provides a good overview of related work but lacks a discussion on the latest lightweight CNN models for plant disease classification (e.g., MobileNetV3, EfficientNet-Lite, GhostNet). Recent advancements, including MobileNetV3 and EfficientNet-Lite, further optimize computational efficiency. However, these models still require more parameters than the proposed GAPNet, making them less suitable for resource-constrained applications. In addition, Explicitly highlight the limitations of existing methods and how GAPNet addresses them. The recent studies on optimized CNN architectures used in agriculture (2023-2025) are required.
*How were training, validation, and test splits determined? this point is not clear in the article

Experimental design

The evaluation utilizes multiple metrics, including accuracy, precision, recall, and F1-score. However, to ensure a more comprehensive assessment of the model’s robustness, additional evaluations are necessary. These should include computational efficiency metrics such as inference time, FLOPs, and memory consumption, as well as statistical validation through k-fold cross-validation. Furthermore, testing GAPNet’s performance on real-world datasets with varying lighting conditions and backgrounds would strengthen the reliability of the findings.

Validity of the findings

*The evaluation uses multiple metrics (accuracy, precision, recall, F1-score). However, additional evaluations are needed for robustness.

Additional comments

* There is space to improve the quality of the language.

Reviewer 3 ·

Basic reporting

The manuscript entitled "GAPNet: Single and multi plant leaf disease classification method based on simplified SqueezeNet for grape, apple and
potato plants" proposes Disease classification was performed using 7 different pretrained convolutional neural networks, namely VGG16, ResNet50,
SqueezeNet, Xception, ShuffleNet, DenseNet121 and MobileNetV2. The authors proposed a simplified SqueezeNet model, called GAPNet, for grape, apple and potato leaf disease classification as a light and fast model.

The contributions of the manuscript are stated point-by-point. Introduction section is satisfactory. Literature Review section can be improved. The method is explained in detail, the experimental results are good and compared with several methods. The references are mostly up-to-date. However, some corrections and improvements are needed for the publication of the manuscript as follows:

1. The authors should cite their own conference paper and state the differences between the submitted manuscript and their conference paper:
"AppleSENet: Apple Leaf Disease Classification Using Lite SE-Network," 2024 8th International Artificial Intelligence and Data Processing Symposium (IDAP), Malatya, Turkiye, 2024

2. Literature review section should be improved by adding the most recent apple, potato and grape leaf disease classification articles published in 2025 or 2024. The following examples should be added into that section:
- A systematic review of deep learning techniques for apple leaf diseases classification and detection, PeerJ Computer Science, 2025
- Apple Leaf Disease Identification: A Hybrid Multi-Scale Deep Learning Model, Journal of Information Systems Engineering & Management, 2025
- Potato leaf disease classification using fusion of multiple color spaces with weighted majority voting on deep learning architectures. Multimedia Tools and Applications, published online: 2024
- Advancements in deep learning for accurate classification of grape leaves and diagnosis of grape diseases. J Plant Dis Prot 131, 1061–1080, 2024

3. Computation time of the proposed method should be mentioned. Computation time of the proposed method should be compared with the computation times of other methods or individual methods (if available).

Experimental design

No comment.

Validity of the findings

No comment.

Additional comments

The manuscript should be improved according to the reviewer's suggestions.

---

## Round 0.2 · accepted · Accept

The author has addressed the reviewers' comments properly. Thus I recommend publication of the manuscript.

Reviewer 1 ·

Basic reporting

acceptable

Experimental design

acceptable

Validity of the findings

acceptable

Additional comments

After revision, this paper is better and acceptable

Reviewer 2 ·

Basic reporting

I am satisfied with the latest revision

Experimental design

No comments

Validity of the findings

No comments

Reviewer 3 ·

Basic reporting

The manuscript entitled "GAPNet: Single and multi plant leaf disease classification method based on simplified SqueezeNet for grape, apple and potato plants" proposes Disease classification was performed using 7 different pretrained convolutional neural networks, namely VGG16, ResNet50,
SqueezeNet, Xception, ShuffleNet, DenseNet121 and MobileNetV2. The authors proposed a simplified SqueezeNet model, called GAPNet, for grape, apple and potato leaf disease classification as a light and fast model.

The contributions of the manuscript are stated point-by-point. Introduction section is satisfactory. Literature Review section is improved in the revised paper. The method is explained in detail, the experimental results are good and compared with several methods. The references are expanded and up-to-date in the revised version.

The authors revised the paper and improved the paper according to reviewer comments. Therefore, the revised version is acceptable.

Experimental design

The experimental results are good and compared with several methods.

Validity of the findings

The experimental results are compared with several methods which show the validity of the findings.

Additional comments

The revised paper is improved and it is acceptable.